# Pharmacotherapy for Pulmonary Hypertension in Infants with Bronchopulmonary Dysplasia: Past, Present, and Future

**DOI:** 10.3390/ph16040503

**Published:** 2023-03-28

**Authors:** Candice D. Fike, Judy L. Aschner

**Affiliations:** 1Department of Pediatrics, University of Utah Health, Salt Lake City, UT 84108, USA; 2Department of Pediatrics, Joseph M. Sanzari Children’s Hospital at Hackensack University Medical Center, Hackensack, NJ 07601, USA; 3Department of Pediatrics, Hackensack Meridian School of Medicine, Nutley, NJ 07110, USA

**Keywords:** chronic lung disease of prematurity, neonatal pulmonary hypertension, sildenafil, L-citrulline

## Abstract

Approximately 8–42% of premature infants with chronic lung disease of prematurity, bronchopulmonary dysplasia (BPD), develop pulmonary hypertension (PH). Infants with BPD-PH carry alarmingly high mortality rates of up to 47%. Effective PH-targeted pharmacotherapies are desperately needed for these infants. Although many PH-targeted pharmacotherapies are commonly used to treat BPD-PH, all current use is off-label. Moreover, all current recommendations for the use of any PH-targeted therapy in infants with BPD-PH are based on expert opinion and consensus statements. Randomized Control Trials (RCTs) are needed to determine the efficacy of PH-targeted treatments in premature infants with or at risk of BPD-PH. Prior to performing efficacy RCTs, studies need to be conducted to obtain pharmacokinetic, pharmacodynamic, and safety data for any pharmacotherapy used in this understudied and fragile patient population. This review will discuss current and needed treatment strategies, identify knowledge deficits, and delineate both challenges to be overcome and approaches to be taken to develop effective PH-targeted pharmacotherapies that will improve outcomes for premature infants with or at risk of developing BPD-PH.

## 1. Introduction

Pulmonary hypertension (PH) is a devastating disorder that can affect patients of all ages, from newborns in the first few hours of life to adults in late life stages. For many years, adult caregivers have utilized registries and a comprehensive PH classification system to assist in the care of their patients with PH [1,2]. Unfortunately, the classification of pediatric PH into distinct subtypes/categories/phenotypes received little attention until the last decade [3,4,5]. As a consequence of the failure to make clear distinctions between phenotypes of PH presenting in the pediatric age group, the approach to treatment for virtually all forms of pediatric PH is largely the same. As with adults with PH [1,6,7,8], it is now appreciated that management strategies for infants and children with PH may need to differ dependent on the endotype, i.e., the underlying pathophysiology and disease-specific features. In particular, treatments that are efficacious for PH that presents acutely in newborns at birth, commonly called persistent pulmonary hypertension of the newborn, PPHN, may not be the optimal treatment for PH presenting in pediatric patients beyond the first few weeks of life [9].

Only recently has PPHN been clearly distinguished from types of neonatal PH that develop postnatally [5,10]. PPHN occurs from the failure to transition from the high fetal pulmonary vascular resistance (PVR) to the lower PVR of the normal newborn lung (Figure 1). This condition frequently complicates acute respiratory failure in the first few hours or days of life [11]. By comparison, many infants who develop PH in later postnatal life undergo an initial normal pulmonary circulatory adaptation in the first hours and days of life and subsequently experience adverse, injurious conditions, such as exposure to prolonged respiratory support or oxidative injury from high FiO_2_, that cause abnormalities in the development of alveoli and pulmonary vasculature that underlie the postnatal development of PH (Figure 1). A postnatal form of developmental PH can occur in premature infants who develop PH associated with chronic lung disease of prematurity, bronchopulmonary dysplasia (BPD, Figure 1) [5]. As the underpinnings of PH associated with BPD (BPD-PH) is quite distinct from the pathophysiology of PPHN, it is understandable that the approach to treatment for these distinct types of neonatal PH might need to be different.

There is no question that research efforts are needed to develop therapies and improve outcomes for neonates with both PPHN and BPD-PH. Mortality for PPHN is approximately 4–33% [12], while BPD-PH continues to carry mortality estimates of up to 47% [13]. Historically, BPD-PH has received less clinical and basic science attention than PPHN. For example, based on research from basic science laboratories, randomized clinical trials (RCTs) were performed in newborns who presented at birth with respiratory failure and PPHN and led to the approval of inhaled nitric oxide (iNO) in late 1999 for the treatment of PPHN in newborns at 35 weeks gestation and older. To date, no RCTs have been performed to evaluate the efficacy of iNO or any other pharmacotherapy to improve the outcome for infants with established BPD-PH. A recent report from the Pediatric Pulmonary Hypertension Network (PPHNet) noted the striking contribution of BPD-PH to the spectrum of pediatric PH and highlighted the need for focused research and RCTs in this understudied group of patients [14]. This review will discuss pharmacotherapeutic strategies for treating BPD-PH. Attention will be given to delineating the pathophysiologic basis for current and future strategies. The major intent is to delineate our knowledge deficits and identify needed approaches to optimize outcomes.

## 2. General Management

Strategies for treating PH in any age group should not be limited to the use of PH-targeted drugs but should incorporate supportive, non-pharmaceutical-based management approaches. For example, supplemental oxygen therapy is recommended as a treatment for infants with BPD-PH with the goal of minimizing exposure of the pulmonary circulation to episodic and sustained periods of hypoxia [15]. This recommendation is based on the tenet that intermittent and prolonged periods of hypoxia are the most likely causes of PH in chronic lung diseases, including BPD [16].

The mechanistic underpinnings by which hypoxia causes PH and the time course of hypoxia-induced PH development have been extensively studied and carefully delineated in a number of experimental animal models, including models in newborn animals [17,18]. During the initial minutes to hours of hypoxic exposure, a reactive, vasoconstrictor response increases pulmonary vascular resistance, which is largely reversible with vasodilators. With extended duration of hypoxia, pulmonary arterial walls thicken, and contribute to a more fixed elevation of pulmonary vascular resistance, with poorer responses to vasodilators [17,19]. Based on the well described adverse consequences of hypoxic exposure on the pulmonary circulation, it is recommended that human infants with BPD-PH avoid oxygen saturations of less than 92% [15]. Of note, O_2_ saturations greater than 95% are also to be avoided because hyperoxia is an oxidant stress that injures the lung parenchyma and pulmonary vasculature of newborns of many species, including humans [20,21]. Paradoxically, but similar to the impact from prolonged hypoxia, exposure to chronic hyperoxia causes pulmonary vascular remodeling and impairs alveolar development in experimental animal models in a fashion similar to that found in human premature infants who develop BPD [20].

Additional supportive measures recommended for infants with BPD-PH include aggressive evaluation for potential aggravators that could worsen the underlying lung disease, such as chronic aspiration and gastroesophageal reflux [15]. Respiratory support should be maximized, ideally avoiding barotrauma with use of non-invasive respiratory support rather than invasive mechanical ventilation [15]. In addition, thorough examination of the airway for structural abnormalities, including tracheomalacia, that can contribute to hypoxemia and poor clinical responses to oxygen, should be performed [15]. Taken together, the major goals of these supportive strategies are to minimize the amount of injurious stimuli to which the lung parenchyma and pulmonary circulation are exposed and to ensure adequate but not excessive oxygenation [15]. Once supportive measures are optimized, consideration can be given to starting PH-targeted pharmacotherapy [15].

## 3. Pharmacotherapy Targeting NO Signaling

The current recommendation is that initiation of PH-targeted pharmacotherapies be considered for those infants with BPD who have evidence of sustained PH after optimizing supportive measures [15]. The PH-targeted therapies that are most often considered for use in infants with BPD-PH are those that utilize the NO signaling pathway [9,13,15,22,23,24,25]. NO is a potent pulmonary vasodilator that is endogenously produced by a variety of cell types including pulmonary vascular endothelial cells and airway epithelial cells [26,27]. NO freely diffuses into smooth muscle cells and activates intracellular soluble guanylate cyclase (sGC) to produce 3′5′-cyclic guanosine monophosphate (cyclic GMP), which mediates most of the physiological effects of NO, including smooth muscle relaxation [26,27].

### 3.1. Inhaled NO

Therapeutically, NO can be administered exogenously as inhaled Nitric Oxide (iNO) gas. At least some infants with BPD have an acute pulmonary vasodilator response to iNO [28,29]. Unfortunately, although some patients appear to clinically improve, at least temporarily, the response to iNO in infants with documented PH and BPD is quite variable [30,31]. Since no RCT has been performed and the amount of published data is so scarce, definitive conclusions about when and how to use iNO in infants with BPD-PH cannot be made (Table 1). Nonetheless, the American Heart Association and the American Thoracic Society included a statement in their 2015 treatment guidelines for Pediatric PH that iNO can be an effective pharmacotherapy for infants with BPD and symptomatic PH [15]. A subsequent guideline published by the PPHNet in 2017 more specifically recommended that iNO be used for an acute PH crisis in infants with BPD-PH and that the iNO be weaned after stabilization [32]. All guidelines have been careful to mention that no RCT has been performed to provide evidence that either acute or prolonged iNO improves the outcome of infants with BPD-PH. All current published recommendations for using iNO in infants with BPD-PH are based on expert opinion and consensus statements [9,13,15,23,24].

### 3.2. Sildenafil

Therapies alternative to iNO that can be used to increase cyclic GMP in the pulmonary vasculature include phosphodiesterase 5 inhibitors, such as sildenafil, that inhibit the breakdown of cyclic GMP [27]. In 2005, both the FDA and the European Union approved oral sildenafil for the treatment of PH in adults. These approvals were based on a large multinational RCT which found that a number of functional and hemodynamic outcomes were improved in adult patients with PH treated with sildenafil [33].

Subsequent to approval as a treatment for adult PH, an RCT, Sildenafil Citrate in Treatment-Naïve Children with Pulmonary Arterial Hypertension (STARTS-1), was performed. STARTS-1 evaluated the dose-ranging impact of oral sildenafil monotherapy in pediatric patients, aged 1–17 years, with a variety of forms of PH, including idiopathic PH, and PH associated with connective tissue disease and congenital heart disease [34]. Although the study found no significant impact on any measured outcome in children treated with a low (10 mg) dose of sildenafil, children receiving a medium (10–40 mg) or high (20–80 mg) dose of sildenafil had improvements in a number of outcomes, including mean pulmonary arterial pressure, and pulmonary vascular resistance index [34]. In the long-term extension study (STARTS-2), favorable survival rates were found in all three sildenafil dose groups [35]. However, an unexplained increased mortality was found in children randomized to high compared to low sildenafil doses [35]. Based on disparate interpretations of the results of these studies, the European Medicines Agency approved the use of specific doses of sildenafil for treating PH in pediatric patients (ages 1–17 years old), whereas the FDA released a warning against the use of sildenafil for the same population of pediatric patients. Consequently, the use of sildenafil in the US to treat PH in children ages 1–17 years old, regardless of type/cause of PH, is off-label and continues to be the subject of scrutiny and controversy [36,37].

It should be noted that the use of sildenafil in patients < 1 year of age is also off-label (Table 1). In fact, no large RCT evaluating sildenafil has yet been performed in infants < 1 year of age, including term infants with PPHN or premature infants with or at risk of developing BPD-PH. An open label dose-escalation study in 36 term and near-term infants with PPHN found improvements in oxygenation with some doses of intravenous sildenafil [38]. Results of some uncontrolled observational studies with small patient numbers have suggested that sildenafil-treatment can have favorable effects and outcomes in patients with BPD-PH [37,39,40,41,42], while findings from some other observational studies should raise concern about the effectiveness of sildenafil as a treatment for BPD-PH. Specifically, one study found that the acute reductions in right ventricular peak systolic pressure that occurred with sildenafil treatment were not accompanied by improvements in gas exchange [43]. In another study, although a majority of sildenafil-treated patients with BPD-PH showed echocardiographic improvement, a minority, only 35%, showed clinical signs of improvement [44]. Despite the lack of robust and consistent data supporting its use, virtually all current published recommendations for treating BPD-PH mention using sildenafil [9,13,15,32,45]. As a result, even though the efficacy and safety of the therapy remain unknown, the use of off-label sildenafil in premature infants with or at risk of developing BPD-PH is very common [46,47], especially in countries or hospitals without access to iNO or other more expensive pharmacologic agents.

The need for an RCT evaluating the efficacy and safety of sildenafil in the treatment of BPD-PH is increasingly acknowledged [46,48,49]. Current FDA guidance indicates that PK studies be conducted to establish the correct dose prior to evaluating the efficacy of any drug in a pediatric population [50]. This FDA guidance is based on the awareness that PK data obtained in adults and older children do not adequately characterize the PK of infants. Moreover, developmental changes in the pediatric population must also be considered, such that PK studies performed in term newborns [51] should not be assumed to represent the PK parameters of preterm infants, the patient population that develops BPD. Thus, to guide the doses and treatment intervals that should be used in an RCT, data are needed on the pharmacokinetics and pharmacodynamics of sildenafil in premature infants with or at risk of developing BPD-PH.

Notably, a study (ClinicalTrials.gov Identifier: NCT01670136) with the primary goal of characterizing the PK profile of sildenafil in premature infants has recently been completed (Table 1) [52,53]. A total of 34 infants were enrolled in the study. A total of 23 adverse events were reported in 15 infants. Only one adverse event, a serious adverse event involving hypotension, was related to the study drug, sildenafil. Results of this phase I study have been used to develop a population PK (PopPK) model and to derive the sildenafil dosing strategies that are currently being evaluated for safety in a phase II trial performed in premature infants at risk of BPD (ClinicalTrials.gov Identifier: NCT03142568). A separate phase II trial is underway with the primary goal of evaluating the safety of sildenafil in premature infants with severe BPD (ClincalTrials.gov Identifier: NCT04447989). Although the primary goal is to determine safety, both of these phase II studies are designed with the intent to collect data to further characterize PK parameters of sildenafil in premature infants. In addition, both studies contain clinical endpoints to provide some preliminary efficacy data. When available, results of the foregoing studies should provide the information needed to guide the choice of optimal dosing used in phase III trials designed to evaluate the efficacy of sildenafil as a treatment for BPD-PH in premature infants.

### 3.3. L-Citrulline

Another way to manipulate the NO signaling pathway as a potential treatment for PH is to give L-citrulline, the amino acid precursor of the NO substrate, L-arginine [54]. Via a two-step enzymatic pathway, L-citrulline boosts intracellular production of L-arginine and thereby enhances the amount of NO produced by pulmonary vascular endothelial cells [55]. Orally or enterally administered L-citrulline has been shown to increase metrics of NO production in children [56] and adults [57] as well as in a newborn piglet model of chronic hypoxia-induced PH [58,59]. Due to interest in using L-citrulline as a pharmaconutrient that can increase NO production and improve endothelium-dependent responses, studies delineating the PK profile of orally administered L-citrulline have been performed in healthy adults [60,61].

No large RCTs have yet been performed in adults or any other age group evaluating the effectiveness of L-citrulline as a treatment for any type of PH. Although not an RCT, oral administration of L-citrulline was shown to reduce pulmonary arterial pressure and increase exercise tolerance in 25 adult patients with either idiopathic pulmonary hypertension or congenital heart disease and Eisenmenger Syndrome [62]. In a small RCT, oral L-citrulline or placebo was given to 40 children < 6 years of age with cardiovascular disease undergoing cardiopulmonary bypass which placed them at risk of developing post-operative PH [63]. Patients with either naturally high baseline concentrations of L-citrulline or those who achieved plasma concentrations exceeding 37 micromolar did not develop post-operative PH [63]. A subsequent study determined the PK profile of intravenously administered L-Citrulline in the same pediatric patient population [64]. Both a phase Ib/II (ClincalTrials.gov Identifier: NCT01120964) and phase III (ClincalTrials.gov Identifier: NCT02891837) RCT have now been performed in which either placebo or intravenous L-citrulline was given to children < 18 years of age undergoing cardiopulmonary bypass during surgery for congenital heart defects. Although neither of these latter two studies include PH as an outcome measure, multiple clinical parameters reflecting development of PH as a complication that lengthens post-operative recovery, e.g., length of time on mechanical ventilation or requiring ionotropic support, are included as either primary or secondary outcomes. Publication of the results of these studies is awaited and should provide valuable information about the safety and effectiveness of IV L-citrulline as a therapy to improve clinical outcomes in children < 18 years of age with congenital heart disease who undergo cardiopulmonary bypass during surgery and are at risk of postoperative PH.

Proof of concept to evaluate enterally administered L-citrulline as a PH-targeted therapy in newborns is provided by studies with the newborn piglet model showing that, in addition to increasing pulmonary vascular NO production, oral L-citrulline treatment inhibited both the onset and progressive development of chronic hypoxia-induced PH [58,59]. Additional rationale to evaluate L-citrulline as a PH-targeted therapy for neonates with BPD comes from a study showing that plasma L-citrulline concentrations were lower in human neonates with BPD-PH than in neonates with BPD and no PH [65]. In accordance with FDA guidelines, rather than relying on results from studies performed in adults receiving enteral L-citrulline [60,61] or older children receiving IV L-citrulline [64], PK studies are needed to guide the doses and treatment intervals that could be used in a RCT to evaluate oral/enteral L-citrulline as an efficacious treatment for premature neonates with BPD-PH. With that goal in mind, a PK study in which a single enteral dose of L-citrulline was given to premature infants at risk of developing BPD-PH was recently published (Table 1) [66]. Ten premature infants were administered a single 150 mg/kg dose of L-citrulline. No patients experienced an adverse event. The data from that single dose study were used to develop a popPK model and to derive an optimal dose of L-citrulline to be given in a multi-dose PK study (ClincalTrials.gov Identifier: NCT03542812). Results of the multi-dose study are pending and will be used to further refine the popPK model of oral L-citrulline in premature infants at risk of developing BPD-PH. The next step will be to perform a phase II study to evaluate the safety of L-citrulline in this premature infant patient population and to provide some pharmacodynamic information and evidence of potential efficacy. When completed, and if the oral citrulline therapy is shown to be safe, these phase I and phase II study findings will provide the information needed to design a phase III study to evaluate the effectiveness of oral L-citrulline as a treatment for premature infants with or at risk of developing BPD-PH.

Of interest, supporting the likelihood that oral citrulline will be a safe therapy to use in premature infants, results of a pilot study that is not yet published but that has been completed (ClinicalTrials.gov Identifier: NCT03649932) lists no adverse effects in 40 premature infants treated with oral citrulline doses of either 100 mg/kg/d (14 infants), 200 mg/kg/d (13 infants), or 300 mg/kg/d (13 infants). Also of interest, a study that intends to provide some safety and PK data about the use of one of two oral L-citrulline dosing regimens (either 300 or 500 mg/kg/d divided q6 h) to treat preterm infants with established BPD (with or without PH) will start recruiting patients in early 2023 (ClinicalTrials.gov Identifier: NCT05636397).

### 3.4. Tadalafil

Tadalafil is a PDE5 inhibitor that has a longer half-life and longer duration of therapeutic effects in adults than sildenafil [67]. Tadalafil was approved for treating adult patients with PH in 2009 after an RCT showed that treatment with tadalafil improved exercise ability and quality of life and delayed time to clinical worsening in adults with PH [68]. A number of observational studies have provided evidence that oral tadalafil is well tolerated and has an acceptable safety profile in pediatric patients with PH age 2 months–18 years [69,70,71,72]. Findings of a phase III RCT that was limited to 35 patients due to challenging recruitment issues provided suggestive but not definitive data that tadalafil is an effective treatment for PH in pediatric patients < 18 years of age [73]. A popPK model has recently been developed that can be used to guide dosing regimens to be used in a future phase III trial evaluating the effectiveness of tadalafil as a treatment for PH in pediatric patients aged 2 to 18 years [74]. To date, there is no information about the PK, safety, or efficacy of tadalafil in premature newborns at risk of BPD-PH.

### 3.5. Riociguat

Other therapies which affect the NO signaling pathway are soluble guanylate cyclase stimulators, such as Riociguat, that increase cyclic GMP independent of NO. Following phase II and phase III trials showing safety and efficacy, Riociguat was approved in 2013 for the treatment of adults with some forms of PH, including chronic thromboembolic PH [75,76]. A study evaluating the safety, tolerability, PK, and potential efficacy of Riociguat for treating PH in children, ages 6–17 years old, is underway (ClincalTrials.gov Identifier: NCT02562235). No studies have been published or appear to be in progress to determine the PK, safety, or efficacy of Riociguat in the term or premature newborn population.

## 4. Pharmacotherapy Targeting Non-NO Signaling Pathways

### 4.1. Endothelin-1 Signaling

Therapies that target endothelin-1 (ET-1) are commonly considered for use in infants with BPD-PH [9,13,23,24,25]. ET-1 has complex pulmonary vasoactive effects that are mediated by at least two distinct receptors, ET_A_ and ET_B_ [77]. Activation of ET_A_ causes vasoconstriction, while ET_B_ activation stimulates NO release and consequent vasodilation [77]. Via ET_A_ activation, ET-1 also stimulates smooth muscle cell growth and can potentiate pulmonary vascular wall remodeling [77].

Bosentan is an orally active non-selective dual endothelin receptor antagonist that was shown to improve exercise capacity, quality of life, and hemodynamics in adult patients with PH [78,79]. After receiving FDA approval in 2001 to treat adult PH, an open-label study was performed that evaluated the PK, safety, and efficacy of bosentan to treat PH in 19 pediatric patients; the youngest patient enrolled was 2 years of age [80]. Several other studies have shown that bosentan is safe and well tolerated in children [81,82,83,84]. An extrapolation approach, in which the beneficial effect of bosentan on pulmonary vascular resistance in children [80] was used to correspond to improvements in exercise tolerance in adults [78,79], led to FDA approval in 2017 for the use of bosentan to treat children > 3 years of age for specific types of PH, idiopathic-genetic and congenital heart disease associated PH.

Macitentan is another dual ET-receptor antagonist that has been FDA approved to treat adults with PH based on results of safety and efficacy studies in that patient population [85]. Macitentan was developed by modifying the structure of bosentan to improve efficacy and safety. Because ET_B_ can cause vasodilation by stimulating NO release, a potential drawback of dual ET-receptor antagonists, like bosentan and macitentan, is that they are nonselective inhibitors of both ET_A_ and ET_B_. Consequently, there has been interest in the development of drugs that are ET_A_ selective receptor antagonists, such as ambrisentan. After being shown to improve exercise tolerance and delay clinical worsening [86], ambrisentan was approved by the FDA to treat some forms of PH in adults. There are some limited data about the PK, safety, and efficacy of ambrisentan to treat children with PH [87]. The clinical experience of using macitentan to treat PH in pediatric patients at a single center has been reported [88]. Further data about the use of endothelin receptor antagonists in the pediatric age group may be forthcoming since ClinicalTrials.gov lists a number of trials that are either on-going or that have been completed in children < 17 years of age. Of note, while acknowledging that the recommendation is largely based on experience and data from adult studies, algorithms for PH treatment in children who fail acute vasoreactivity testing commonly include an endothelin receptor antagonist, specifically bosentan or ambrisentan, as a treatment of choice [5,10,45].

No endothelin-receptor antagonist is FDA approved for use in term or pre-term infants with any kind of PH, including BPD-PH (Table 1). Bosentan is associated with liver toxicity so that careful evaluation and confirmation of its safety profile is mandated prior to its wide-spread use in vulnerable patients, like term and premature infants. An RCT evaluating bosentan vs. placebo for the treatment of PPHN in 47 infants ≥ 34 weeks gestation found that those infants randomized to receive bosentan had improvements in both short-term and longer-term outcomes at 6 months of age without evidence of liver toxicity [89]. A more recent RCT evaluated use of bosentan as an adjunctant therapy to iNO in 21 infants > 34 weeks gestational age with PPHN [90]. Results of this latter study showed that, although the drug was well tolerated with no evidence of liver toxicity, infants treated with bosentan had no greater improvement in any clinical outcome measurements than infants receiving placebo [90]. There are reports describing the use of bosentan to treat premature infants with BPD-PH [40,91]. However, there is no PK, safety, or efficacy data available to guide the use of bosentan in premature infants, including those with BPD-PH. Despite the lack of data in premature infants and the presence of very limited safety data and inconsistent efficacy data in newborns > 34 weeks with PPHN, bosentan therapy is often mentioned as a potential beneficial therapy for PPHN [9,45] for persistent PH in infants with congenital diaphragmatic hernia [92] and for BPD-PH [9,13,32].

### 4.2. Prostacyclin Signaling

Therapies that utilize prostacyclin (PGI_2_) signaling are another category of targeted therapies that do not directly increase NO signaling commonly considered for use in infants with BPD-PH [9,13,23,24,25]. PGI_2_, also called prostaglandin I_2_, is a potent pulmonary and systemic vasodilator produced from prostaglandin H_2_ via prostacyclin synthase in endothelial cells [93,94,95]. PGI_2_ mediates vasodilation by activating adenyl cyclase and increasing intracellular cyclic adenosine monophosate (cAMP) in smooth muscle cells [93,94,95]. In addition to its profound vasodilatory properties, PGI_2_ inhibits cell proliferation and can protect against pulmonary remodeling [93,94,95].

Epoprostenol is a synthetic version of PGI_2_ that requires continuous intravenous administration. In the 1990s, epoprostenol was shown to improve hemodynamic measurements, clinical symptoms, and survival in adults with PH [96]. In 1995, epoprostenol became the first drug to be approved by the FDA for the treatment of PH in adults. Subsequently, two additional PGI_2_ analogs, trepostinil and iloprost, were developed, shown to be efficacious [97,98], and received FDA approval for treating certain types of PH in adults. A significant practical problem posed by the aforementioned drugs is that they require an invasive and/or cumbersome route of administration [99]. Specifically, they must be administered as an intravenous (epoprostenol, treprostinil, iloprost), subcutaneous (treprostinil), or inhalation (iloprost and treprostinil) therapy [99]. To address the challenges associated with these problematic delivery modes, oral therapies targeting PGI_2_ signaling have been developed. Specifically, an oral form of treprostinil [100,101,102] and an oral selective-agonist of PGI_2_ receptors, selexipag [103], were developed, evaluated for efficacy, and approved by the FDA for treating certain forms of adult PH. Beraprost [104] is an oral PGI_2_ analog that was approved for treating PH in adults in Japan and South Korea, but, due to a high incidence of adverse events and concern with evidence of waning efficacy over time, it was not approved in the US [105,106].

None of the drugs targeting PGI_2_ signaling are approved for treating any form of PH in any pediatric age group. Although no large RCTs have been performed, a few studies have been performed in children that have reported findings supporting the potential efficacy of PGI_2_ analogs for treating children with PH. Long-term intravenous treatment with eproprostenol improved the 4-year survival rate for children with primary pulmonary hypertension [107] and improved hemodynamics and the quality of life in children with severe PH associated with congenital heart disease [108]. When added to background PH therapy, inhaled treprostenil was associated with improved exercise tolerance in children with PH [109]. More definitive information about use of treprostenil in the pediatric age group may be forthcoming as a study evaluating the safety, tolerability, and PK of oral treprostinil in children with PH, ages 7–17 years old, is listed as having been completed (ClinicalTrials.gov Identifier: NCT02276872). Another study evaluating the PK of subcutaneous and intravenous treprostinil in children with PH, ages 0–16 months old, is also listed as having been completed (ClinicalTrials.gov Identifier: NCT02318186). Results from these trials are not yet published.

There is very little data about the use of any PGI_2_ signaling drug in newborns. In a retrospective review, thirty-five infants diagnosed with PPHN were reported to show improvements in some clinical parameters when treated with intravenous iloprost [110]. A couple of other studies have reported clinical improvements in term [111,112,113,114] or preterm [115] infants with PPHN treated with inhaled iloprost or inhaled epoprostenol [116]. A retrospective study reported that 7 infants with PPHN had improvements in oxygenation index when treated with beraprost [117]. Clinical improvements have also been described in infants with PPHN treated with epoprostenol [118]. Moreover, two preterm infants with PPHN were reported to improve clinically when treated with treprostinil [119]. An RCT comparing the use of intravenous treprostinil vs. placebo as an adjunct to iNO is currently recruiting infants ≥ 34 weeks gestation with PPHN and aims to provide some efficacy, safety, dosing, and PK information about treprostinil therapy in that neonatal population (ClinicalTrials.gov Identifier: NCT02261883). Additional RCTs are needed to guide the use of prostacyclin signaling drugs in newborns with PPHN.

As to the population of premature newborns with BPD-PH, a few case reports have described clinical improvements when using either intravenous epoprostenol [120,121], subcutaneous treprostinil [122], or inhaled iloprost in premature infants with BPD-PH [123,124,125]. A retrospective study has been published describing the clinical experience at one institution for treating premature infants with BPD-PH with inhaled iloprost (35 patients), intravenous epoprostenol (12 patients), or subcutaneous treprostinil (9 patients) [91]. Other than these few descriptive reports, there is no PK, safety, or efficacy data available for of any of the drugs that target PGI_2_ signaling in premature infants with BPD-PH (Table). Nonetheless, numerous drugs that target PGI_2_ signaling are often cited as being beneficial for treating infants with BPD-PH [9,13,32].

## 5. Challenges for RCT Design in Patients with or at Risk of Developing BPD-PH

The need to acquire the PK and safety data to inform the choice of optimal dosing strategies is not the only challenge facing the design of RCTs to evaluate the efficacy of sildenafil, L-citrulline, or any other therapy in infants with BPD-PH. Clinical trial endpoints and definitions are among the other significant study design challenges in this patient population.

### 5.1. Therapeutic Goals

One issue is that the therapeutic goals of PH targeted treatments for premature infants need careful consideration and re-examination. Traditionally, the primary goal of PH therapy in newborns has been to elicit pulmonary vasodilation [126,127]. This is certainly the therapeutic need for infants with PPHN [126,127] and is also an appropriate primary therapeutic goal for infants with BPD-PH who are experiencing an acute PH crisis [32,128]. However, by itself, pulmonary vasodilation is an insufficient treatment goal for a developmental disorder such as BPD-PH. Premature newborns are born at a critical time in lung development during which future alveoli and distal pulmonary vasculature would be rapidly forming had the fetus remained in utero. When exposed to injurious conditions for long periods of time, such as exposure to positive pressure respiratory support and supplemental oxygen for days to months, the alveoli and pulmonary vasculature fail to develop normally, both structurally and functionally [21,129,130,131]. Structural abnormalities include pulmonary vascular wall thickening (Figure 1) and failure to develop the distal pulmonary circulation [21,129,130,131]. Functional impairments include exaggerated vasoconstrictor responses and a reduced ability to vasodilate [21,129,130,131]. Taken together, these abnormalities severely reduce the total cross-sectional area of the pulmonary circulation. To address all the functional and structural abnormalities in pulmonary vascular development, therapeutic goals should not be limited to eliciting vasodilation, but should include reversing structural changes in the vascular walls and promoting growth of new vessels that function normally [21,129,130,131].

### 5.2. Eligibility Criteria

The current recommendation is for PH-targeted pharmacotherapies to be initiated in those infants with BPD who have evidence of sustained PH [15]. However, starting therapy after PH is diagnosed and well established has inherent limitations. It has been shown in adults that a rise in resting pulmonary arterial pressure is a late event in the natural history of pulmonary vascular disease because of microvasculature reserves. Specifically, resting pulmonary arterial pressure rises only when the distal pulmonary circulation has been reduced by ≥50% [132]. Thus, at the time that PH can be detected as an increase in pulmonary arterial pressure by transthoracic echocardiography (TTE) or cardiac catheterization, the damage to the pulmonary vasculature is quite extensive and the ability to reverse adverse changes may not be possible or will take a very long time. In a developmental disorder like BPD-PH, the damage to existing pulmonary vessels that takes place during the evolution of PH is compounded by the fact that the growth of new, normal vessels will be impaired and/or fail to occur [21,129,130,131,133]. It seems logical that, to have a maximal impact on promoting normal vascular growth that must take place in the developing lung, consideration should be given to initiating therapies in RCTs as early as seems safe and practical. In other words, RCTs that evaluate the impact of therapies should be started in infants with BPD-PH prior to the point at which significant PH is detectable by TTE or catheterization [134], i.e., RCTs with the goal of preventing, not just decreasing the severity of existing BPD-PH. RCTs should be designed to evaluate the efficacy of starting therapy in premature infants with known risk factors for BPD-PH development.

Lack of criteria that accurately identify infants with the highest risk of developing BPD-PH hampers the design and eligibility criteria for prevention-based RCTs of novel therapeutic approaches. Results of studies have shown that the incidence of PH development correlates with the severity of BPD, i.e., infants with severe BPD have a higher incidence of PH than those with mild or moderate BPD [135,136]. Thus, one suggested approach is to target eligibility to patients with severe BPD so that only infants with BPD at the highest risk of PH are studied. One limitation of this approach is that currently the diagnosis of BPD is made at 36 weeks postmenstrual age (PMA) [137,138,139]. Waiting until 36 weeks PMA to start therapy in an RCT means that the premature lung and pulmonary vasculature are likely to have been exposed to a prolonged period of injurious stimuli such that the damage will be difficult or impossible to reverse. In other words, waiting until 36 weeks PMA to start therapy means that the ability to elicit structural and functional improvements in the pulmonary vasculature may take months of therapy and be difficult to detect.

An additional approach would be to design an RCT in which therapy is started in premature infants at a timepoint before a formal BPD diagnosis is made. To minimize the number of infants unnecessarily exposed to potential toxicity from a therapy with unknown efficacy, eligibility should include criteria that indicate the infant is at high risk of disease development. For example, preterm infants with prolonged premature rupture of membranes and/or early PPHN diagnosed in the first postnatal week of life have been shown to be at increased risk of late PH [9,136,140,141]. This may be an important subgroup of preterm infants to target in a clinical trial of an experimental therapeutic intervention. An additional approach that holds merit would be to study infants born at early gestational ages who have required a postnatal level of respiratory support for a significant duration or oxygen requirement that place them at high risk of developing BPD. For example, eligibility criteria chosen for the recently published single dose L-citrulline PK study were infants born at ≤28 weeks gestation and requiring invasive, mechanical ventilation or non-invasive positive pressure support (nasal continuous positive airway pressure) or high flow nasal cannula ≥ 1 L/min) at 32 ± 1 weeks PMA [66]. It cannot be overstated that establishing safety of the treatment in premature newborns is a necessary step prior to performing an efficacy trial in this fragile patient population. Treatments, such as L-citrulline, that are currently taken as a nutritional supplement by adults, and have been used for decades without evidence of toxicity in children with certain types of urea cycle defects [142,143], hold particular promise for being shown to be safe to use in premature patients who have not yet developed BPD.

### 5.3. Therapeutic End-Points

Another challenge that needs to be overcome is the paucity of validated endpoints for assessing clinical course and responses to therapy in neonates. Efficacy end-points should be clinically relevant, sensitive to treatment effect, measurable, interpretable, and achievable [50,144].

#### 5.3.1. Transthoracic Echocardiographic (TTE) End-Points

It is extremely problematic that there are well described limitations in the ability of TTE to detect the presence or degree/severity of PH [145,146,147,148]. Hemodynamic measurements obtained during intracardiac catheterization remain the “gold standard” for diagnosing PH. Due to the invasive nature and concern for poor tolerance of the procedure, intracardiac catheterization is not recommended for use as a screening tool to diagnose the presence or severity of PH in premature infants. Hence, even though no evidence-based study or consensus statement is available to guide the choice of TTE metrics to use, TTE is the most practical modality to be used for detection and quantification of PH in RCTs performed in premature infants. Ideally, to be able to enhance biostatistical data analysis and minimize the number of patients studied, rather than using a binary outcome such as presence or absence of PH, a TTE metric that is reported as a continuous, numerical value should be used. Unfortunately, only a few TTE metrics, including estimates of peak systolic pulmonary artery pressure (sPAP) based on the tricuspid regurgitation jet (TRJV), pulmonary artery acceleration time (PAAT), and left ventricular systolic eccentricity index (LV-sEI), are reported as numeric values, and none of them are consistently and reliably measured in all premature infants [149,150]. Moreover, there continues to be concern regarding how accurately any TTE metric reflects measurements of pulmonary arterial pressure obtained during cardiac catheterization [146,151,152,153]. Consequently, the presence and severity of PH cannot be relied upon as the sole end-point to be used in an RCT performed to evaluate treatment efficacy in patients with or at risk of developing BPD-PHD.

#### 5.3.2. Respiratory End-Points

Since PH development correlates with the severity of BPD [135,136], one possibility is to use a respiratory end-point based on the NICHD data-driven diagnostic criteria for severity of BPD. These diagnostic criteria categorize BPD severity based on the mode of respiratory support administered at 36 weeks PMA [138] and provide the best prognostic accuracy for predicting death and serious respiratory morbidity at 18–26 months corrected age [154]. Other respiratory parameters that correlate with severity of BPD, such as days on mechanical ventilation, non-invasive ventilation, and supplementary oxygen, or a numerical respiratory severity score, consisting of the mean airway pressure multiplied by a fraction of inspired oxygen, could be used. An additional approach would be to use clinical worsening as a composite end-point [155].

#### 5.3.3. Long-Term End-Points

All of the above are short-term end points that may or may not be strongly associated with long-term, clinically meaningful outcomes [156]. Outcomes that are meaningful to families and to the child as they grow older are more important from a health and society standpoint than changes in a physiologic parameter or imaging test, particularly if that change is not associated with long-term morbidity, mortality, or quality of life [156]. While more costly and necessitating trials with an outcome that is potentially several years following the intervention, it is important that families who consent to their child’s participation in a clinical trial know that the outcome that is being targeted for improvement is one that is meaningful to them. Outcomes such as need for home technology, need for daily medications, hospitalizations, and Emergency Department visits for respiratory or cardiac reasons, as well as neurodevelopmental status and quality of life outcomes, have all been proposed as composite or individual primary or secondary outcomes [156]. Future trials should also incorporate a plan for communicating trial findings (return of results) to participants and to the public at large.

## 6. Additional Comments about RCTs Manipulating NO Signaling

In addition to being a potent pulmonary vasodilator, NO is known to inhibit vascular wall smooth muscle growth [157,158] and to promote both angiogenesis and alveolarization [159]. Therapies that manipulate NO signaling should therefore be ideal candidates to use to target the functional and structural abnormalities in lung development that are prominent pathologic features in premature infants with BPD-PH. Animal models have demonstrated the biological plausibility that manipulating NO signaling might ameliorate the development of BPD-PH. Reduced NO production was implicated in the pathogenesis of impaired alveolar and vascular growth found in a number of animal models of BPD-PH [160,161,162,163]. iNO treatment improved lung angiogenesis and alveolarization in studies performed with rat models of BPD [159,160,164]. Lung structural and functional abnormalities were improved by long-term iNO treatment in chronically ventilated preterm lambs [165] and baboons [163].

Studies with animal models have also demonstrated that treatment with either sildenafil or L-citrulline can have a beneficial impact on the pulmonary circulation and lungs that extends beyond that of a pulmonary vasodilator. L-citrulline treatment reduced pulmonary vascular wall thickness and increased lung capillary formation in a newborn piglet model of chronic hypoxia-induced PH [166]. Alveolar and vascular growth were preserved [167,168], and pulmonary arterial wall thickness and right ventricular wall hypertrophy were reduced by L-citrulline treatment in a hyperoxic newborn rat model of BPD-PH [168]. In other studies performed with hyperoxic newborn rat models of BPD, sildenafil increased alveolarization and angiogenesis and reduced pulmonary vascular wall thickness and right ventricular hypertrophy [169,170]. In a study with hyperoxic newborn mice, sildenafil treatment reduced right ventricular hypertrophy and pulmonary vascular wall thickness but did not improve alveolarization or angiogenesis [171]. Of note, the difference in findings between hyperoxic rats and mice could be due to use of much higher doses of sildenafil (50–150 mg/kg/d) in studies with hyperoxic rats than in the study with hyperoxic mice (3 mg/kg every other day).

RCTs evaluating therapeutic efficacy of iNO, sildenafil, or L-citrulline to treat or prevent BPD-PH have not been performed in premature human infants. Nonetheless, a number of RCTs have evaluated the ability of prolonged iNO to prevent the development of BPD in this patient population. Despite the strong, consistent evidence provided by studies with various animal models of BPD as noted above, results from RCTs evaluating whether iNO can ameliorate BPD in premature infants have been inconsistent. For example, an early single-site RCT that was performed in premature infants < 34 weeks gestation requiring mechanical ventilation for respiratory distress syndrome in the first 72 h of life, found that the primary outcome of death or BPD was less in the iNO group than in the placebo group [172]. In a subsequent RCT, iNO therapy failed to reduce the outcome of death or BPD in premature infants with severe respiratory failure [173]. Additional RCTs continued to show that, although subsets of patients had evidence of benefit [174], starting iNO in the first few days of life in infants < 34 weeks gestation with respiratory distress fails to ameliorate BPD in the entire cohort of patients [174,175].

The National Institutes of Health convened a consensus panel in October 2010 to evaluate the use of iNO in infants born at less than 34 weeks. After reviewing the published evidence, the panel concluded that use of iNO should be avoided in this group of infants [176]. Subsequently, the Committee on Fetus and Newborn published additional guidance on the use of iNO in this group of infants [177]. In agreement with the 2010 panel, the 2014 panel did not support treating preterm infants with respiratory failure in the first few days of life with iNO to prevent/ameliorate BPD. However, the panel took into consideration the results of an RCT that showed benefits from starting iNO therapy of 20 ppm in premature infants, born at ≤32 weeks gestation with birth weight ≤ 1250 g, still requiring significant respiratory support on days of life 7–21 [178]. That is, based on the multicenter RCT showing that iNO improved the primary outcome of survival without BPD in infants with evolving BPD [178], with the greatest effect in infants enrolled between 7–14 days of life, the 2014 panel stated that treatment with a high dose (20 ppm) of iNO in the second postnatal week may provide a small reduction in the incidence of BPD. They cautioned that the results need to be confirmed by other trials.

The failure of RCTs performed to evaluate the efficacy of iNO to ameliorate BPD in human premature infants to replicate findings with animal models reinforces the need to perform carefully designed RCTs evaluating the efficacy of any treatment for BPD-PH. For example, careful consideration should be given to the time in the evolution of BPD that therapy is initiated and appropriate trial endpoints that are clinically meaningful and important to families.

It should be pointed out that one potential reason that iNO failed to demonstrate a significant impact on BPD is that, in the presence of excess superoxide (O_2_^−^), exogenously administered NO can be inactivated by its interaction with O_2_^−^ to form peroxynitrite (OONO-) [179]. In turn, OONO- can nitrosylate a variety of proteins, including sGC, thereby reducing the ability of NO to stimulate cGMP formation [180]. The complex biochemistry of NO, O_2_^−^, and OONO- and consequent impact on biologic outcomes is likely to differ between human premature infants with respiratory failure and animal models of BPD. It is worth mentioning that, rather than increasing NO by exogenous administration, L-citrulline increases the endogenous intracellular production of arginine which is then metabolized to NO by eNOS [181]. Moreover, sildenafil increases cGMP, not NO. Therefore, neither sildenafil nor L-citrulline are likely to promote excess OONO- formation and its adverse consequences that likely occur with exogenously administered iNO. Nonetheless, RCTs must be performed to determine whether a similar beneficial impact from treatment with sildenafil or L-citrulline in animal models will be found in human premature infants.

## 7. Summary and Conclusions

To date, pharmaceutical studies have focused on adult patients. Treatment for adult PH is evidence-based [6], and there are 14 FDA approved drugs for treating adults with PH [8]. Only 2 drugs, iNO and bosentan, have been approved by the FDA for treating any form of PH in the pediatric age group (Table 1). iNO is approved for treating PPHN in term and near-term infants. Bosentan is FDA authorized for use in treating PH in pediatric patients > 3 years of age. No PH-targeted drug is currently approved for treating premature infants with or at risk of developing BPD-PH (Table 1). Hence, although many PH targeted drugs are used to treat BPD-PH, all current use is off-label. There is a growing awareness that RCTs are needed to evaluate the efficacy of PH targeted treatments in premature infants with or at risk of BPD-PH. Optimal RCT design requires obtaining specific PK, pharmacodynamic (e.g., hemodynamic, clinical endpoints) and safety data. Results of PK, pharmacodynamic, and RCTs performed in adults or other age groups of pediatric patients, including term newborns with PPHN, cannot be extrapolated to premature infants. Studies are now being performed to provide the information needed to optimally design phase III RCTs that will evaluate the efficacy of at least two potential therapies, sildenafil and L-citrulline, to treat preterm infants with or at risk of BPD-PH (Table 1). The efficacy endpoints used in the phase III RCTs require careful thought and should reflect meaningful clinical improvements. Many challenges face pediatricians and neonatologists involved with the design and implementation of phase I, II, and III clinical trials intended to improve the outcomes in such a fragile patient population. Rigorous efforts must continue so that, similar to the progress that has taken place over the past few decades for adults with PH, there will be FDA approved PH-targeted therapies for premature infants with or at risk of BPD-PH in the not too distant future.

## Figures and Tables

**Figure 1 pharmaceuticals-16-00503-f001:**
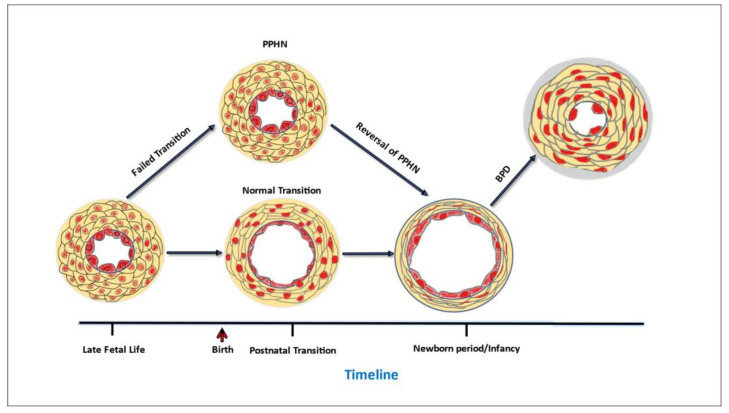
Timeline of critical changes in pulmonary vascular wall structure that occur at key developmental stages.

**Table 1 pharmaceuticals-16-00503-t001:** Status of pulmonary hypertension targeted drugs for treating premature infants with BPD-PH.

Drug Name	Signaling Pathway	Type of Studies Published for Use in BPD-PH	Clinical Trial Phase for Use in BPD-PH	FDA Approval
iNO	NO-cyclic GMP	Observational	No phase 1, 2, or 3 studies currently planned or ongoing for BPD-PH	Not approved for BPD-PH Approved for PPHN in term and near-term newborns
Sildenafil	NO-cyclic GMP	Observational Phase 1 PK study in premature infants at risk of BPD NCT01670136	Phase 2 study in infants at risk of BPD NCT03142568 Phase 2 study in infants with severe BPD NCT04447989	Not approved for BPD-PH Approved for adults with PH
L-citrulline	NO-cyclic GMP	Phase 1 PK study in premature infants at risk of BPD-PH NCT03542812	Phase 1 PK and safety study in premature infants with severe BPD NCT05636397	Not approved for BPD-PH Not approved for any age group of patients with PH
Bosentan	Endothelin-1	Observational	No phase 1, 2, or 3 studies currently planned or ongoing for BPD-PH	Not approved for BPD-PH Approved for adults and children > 3 yrs of age with PH
Epoprostenol	Prostacyclin	Observational	No phase 1, 2, or 3 studies currently planned or ongoing for BPD-PH	Not approved for BPD-PH Approved for adults with PH
Treprostinil	Prostacyclin	Observational	No phase 1, 2, or 3 studies currently planned or ongoing for BPD-PH	Not approved for BPD-PH Approved for adults with PH
Iloprost	Prostacyclin	Observational	No phase 1, 2, or 3 studies currently planned or ongoing for BPD-PH	Not approved for BPD-PH Approved for adults with PH

## Data Availability

Data is contained within the article.

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
