# Peer review of "Pharmacotherapy for Pulmonary Hypertension in Infants with Bronchopulmonary Dysplasia: Past, Present, and Future"

_pharmaceuticals, 2023, doi:10.3390/ph16040503_

Round 1
Reviewer 1 Report
The manuscript is interesting but it not fully ready for a scientific review. The article should focus on the punctuation and use of academic words. Some words are used which are suitable for common English but not for scientific writing. Most of the paragraphs in the whole manuscript have no reference. It is a review and all the statements should include references. It is not authors personal opinion. The authors opinion should be on the conclusion section where they used references. Overall the manuscript needs better organisation and resubmission.
Author Response
Reviewer 1:
The manuscript is interesting but it not fully ready for a scientific review. The article should focus on the punctuation and use of academic words. Some words are used which are suitable for common English but not for scientific writing. Most of the paragraphs in the whole manuscript have no reference. It is a review and all the statements should include references. It is not authors personal opinion. The authors opinion should be on the conclusion section where they used references. Overall the manuscript needs better organisation and resubmission.
Response: All current therapies used to treat BPD-PH are off-label. We have extensively referenced the RCTs in adults and older children that serve, inappropriately, as the basis for treating premature infants with BPD-PH. We also extensively reference the expert opinion and consensus statements which provide the recommendations for treating BPD-PH that are currently clinically used. The statements expressed in the manuscript that are “opinions” are those that are provided by published consensus statements which we reference. We are unable to reference randomized clinical trials in premature infants with BPD-PH because none have yet been performed. Our intent is to delineate both challenges to be overcome and approaches to be taken to develop effective PH-targeted pharmacotherapies that will improve outcomes for premature infants with or at risk of developing BPD-PH. The approaches we discuss are based on current FDA guidance, which we also reference.
Reviewer 2 Report
It is an honor to review your paper.
This manuscript was written to discuss pharmacotherapeutic strategies for treating BPD-PH in the introduction.
In the case of this manuscript, the treatment strategy according to the therapeutic drug was described in the past, present, and future, and the mechanism of the drugs and the current clinical approval status were well described.
Currently, BPD-PH treatment drugs in premature infants have not been FDA-approved, so BPD and PPHN diseases or adult treatment drug studies have been mainly described.
Therefore, it was difficult to say that it focused on the strategy of BPD-PH treatment of premature babies mentioned in the introduction, and this was regrettable.
it has the following revised parts. They should be checked prior to the publication. Followings are recommended for the revision.
Minor revisions
1. Please clarify whether it is Treprostenil or Treprostinil.
2. When explaining the research results, please write in detail how many people were tested and how many of them had no side effects.
3. Check if the sentence above the paragraph "5.0 Challenges for RCT design with or at risk of developing BPD-PH" on page 9 is written correctly.
4. Please make sure that the first sentence of the paragraph "5.2 Eligibility Criteria" is also written correctly.
5. What is the full name of PPROM in page 10?
6. Why did you divide 5.3 into 5.3.1, 5.3.2 and 5.3.3? There was a title for each paragraph, but is it missing?
7. What is the full name of RVH on page 12?
Author Response
Reviewer 2:
This manuscript was written to discuss pharmacotherapeutic strategies for treating BPD-PH in the introduction.
In the case of this manuscript, the treatment strategy according to the therapeutic drug was described in the past, present, and future, and the mechanism of the drugs and the current clinical approval status were well described.
Currently, BPD-PH treatment drugs in premature infants have not been FDA-approved, so BPD and PPHN diseases or adult treatment drug studies have been mainly described.
Therefore, it was difficult to say that it focused on the strategy of BPD-PH treatment of premature babies mentioned in the introduction, and this was regrettable.
it has the following revised parts. They should be checked prior to the publication. Followings are recommended for the revision.
Minor revisions
- Please clarify whether it is Treprostenil or Treprostinil.
Response: All publications that we reference use “treprostinil.”
- When explaining the research results, please write in detail how many people were tested and how many of them had no side effects.
Response: We now detail how many infants were included and how many of them had adverse events recorded for the sildenafil and L-citrulline PK studies that have been completed in premature infants that are at risk of BPD-PH. We have not detailed the number of participants and side effects for the numerous RCTs performed in other ages groups that are cited in the manuscript as they are not informative about the patient population of interest, prematures at risk of BPD-PH, and those details have been provided in other reviews that are relevant to those patient populations.
- Check if the sentence above the paragraph "5.0 Challenges for RCT design with or at risk of developing BPD-PH" on page 9 is written correctly.
Response: We have changed the sentence to read: “Nonetheless, numerous drugs that target PGI2 signaling are often cited as being beneficial for treating infants with BPD-PH.”
- Please make sure that the first sentence of the paragraph "5.2 Eligibility Criteria" is also written correctly.
Response: We have deleted the sentence.
- What is the full name of PPROM in page 10?
Response: We have changed PPROM to prolonged, premature, rupture of membranes.
- Why did you divide 5.3 into 5.3.1, 5.3.2 and 5.3.3? There was a title for each paragraph, but is it missing?
Response: We now provide the titles for each paragraph.
5.3.1 Transthoracic Echocardiographic (TTE) end-points
5.3.2 Respiratory endpoints
5.3.3 Long-term endpoints
- What is the full name of RVH on page 12?
Response: We have changed “RVH” to “right ventricular hypertrophy”.
Reviewer 3 Report
The authors did a great job of addressing such an important topic and they briefly discussed it in a good way.
I caught a typo in the first line of the introduction (afflict---affect)
Author Response
Reviewer 3:
The authors did a great job of addressing such an important topic and they briefly discussed it in a good way.
I caught a typo in the first line of the introduction (afflict---affect)
Response: We thank the reviewer for the support of the manuscript. We have changed “afflict” to “affect”.
Round 2
Reviewer 1 Report
The authors did not put the references appropriately.
Author Response
We have now added References to the manuscript so that many statements that were not previously referenced are now referenced.